# Not All Knowledge Is Created Equal:
# Mutual Distillation of Confident Knowledge

**Ziyun Li**
Hasso Plattner Institute, Germany
ziyun.li@hpi.de

**Xinshao Wang**[*]
University of Oxford, UK
Zenith Ai, UK
xinshaowang@gmail.com

**Di Hu**
Renmin University of China, China
dihu@ruc.edu.cn

**Neil M. Robertson**
Queen's University Belfast, UK
n.robertson@qub.ac.uk

**David A. Clifton**[†]
University of Oxford, UK
davidc@robots.ox.ac.uk

**Christoph Meinel**
Hasso Plattner Institute, Germany
christoph.meinel@hpi.de

**Haojin Yang**
Hasso Plattner Institute, Germany
haojin.yang@hpi.de

## Abstract

Mutual knowledge distillation (MKD) improves a model by distilling knowledge from another model. However, *not all knowledge is certain and correct*, especially under adverse conditions. For example, label noise usually leads to less reliable models due to undesired memorization. Wrong knowledge harms the learning rather than helps it. This problem can be handled by two aspects: (i) knowledge source, improving the reliability of each model (knowledge producer) improving the knowledge source's reliability; (ii) selecting reliable knowledge for distillation. Making a model more reliable is widely studied while selective MKD receives little attention. Therefore, we focus on studying selective MKD and highlight its importance in this work. Concretely, a generic MKD framework, Confident knowledge selection followed by Mutual Distillation (CMD), is designed. The key component of CMD is a generic knowledge selection formulation, making the selection threshold either static (CMD-S) or progressive (CMD-P). Additionally, CMD covers two special cases: zero knowledge and all knowledge, leading to a unified MKD framework. Extensive experiments are present to demonstrate the effectiveness of CMD and thoroughly justify the design of CMD.

---

[*]Work mainly done when being a Postdoc at the University of Oxford.

[†]Prof. David A. Clifton was supported by the NIHR Oxford Biomedical Research Centre, the InnoHK Hong Kong Centre for Cerebro-cardiovascular Health Engineering (COCHE), and the Pandemic Sciences Institute at the University of Oxford.

2022 Trustworthy and Socially Responsible Machine Learning (TSRML 2022) co-located with NeurIPS 2022.

Table 1: The interactions between how each model is trained (i.e., LS, CP, ProSelfLC, and our proposed variant MyLC) and what knowledge should be distilled (zero knowledge, all knowledge, and our proposed CMD-S/P). Experiments are done on CIFAR-100 using ResNet34. The symmetric label noise rate is 40%. The average final test accuracies (%) of two models are reported. The performance difference between the two models is negligible.

| Distilled Knowledge | Label smooth (LS) | Confidence penalty (CP) | ProSelfLC | MyLC |
|---|---|---|---|---|
| Zero | 51.53 | 50.06 | 62.75 | 65.04 |
| All | 53.63 | 53.18 | 59.26 | 61.11 |
| CMD-S (ours) | 55.10 | 53.86 | 67.26 | 68.45 |
| CMD-P (ours) | **56.73** | **56.47** | **68.29** | **69.09** |

# 1 Introduction

"*What knowledge to be selected for distillation*" is an essential question of mutual knowledge distillation (MKD) but has received little attention. Existing MKD methods treat all knowledge of a deep model equally, i.e., all knowledge is distilled into another model without selection. However,

*Should all knowledge or partial knowledge of a model be distilled into another model?*

In clean scenarios, the knowledge source is generally reliable. Thus, simply distilling all knowledge is reasonable, and it has widespread use in existing KD works. However, in label-noise scenarios, the knowledge source is less reliable. The distilled incorrect knowledge would mislead the learning rather than help. Therefore, it is vital to note "not all knowledge is created equal" and identify "what knowledge could be distilled?". We work on this problem from two aspects: (i) *making the knowledge source more reliable*. (ii) *selecting the certain knowledge to distill*. For the first aspect, many algorithms have been proposed, e.g., Tf-KD [1] and ProSelfLC [2]. For simplicity, we exploit them and focus more on the second aspect: *selective knowledge distillation*.

To explore the knowledge selection problem, we design a selective MKD framework, i.e., mutual distillation of confident knowledge, which is shown in Figure 1. We propose to only distill confident knowledge. Specifically, we design a generic knowledge selection formulation, so that we can either fix the knowledge selection threshold *(CMD-Static, shortened as CMD-S)* or change it progressively as the training progresses *(CMD-Progressive, abbreviated as CMD-P)*. In CMD-P, we leverage the training time to adjust how much knowledge would be selected dynamically considering that a model's knowledge improves along with time. CMD-P performs slightly better than CMD-S, according to our empirical studies, e.g., Table 1.

We summarise our contributions as follows:

- We study what knowledge to be selected for distillation in MKD. Correspondingly, we propose a generic knowledge selection formulation, which covers the variants of zero-knowledge, all knowledge, CMD-S, and CMD-P.
- Thorough studies on the models' learning curves, knowledge selection criterion's settings, and hyperparameters justify the rationale of our selective MKD design and its effectiveness.

# 2 Background

KD is an effective method for distilling the knowledge of complex ensembles or a cumbersome model (usually named teacher models) to a small model (usually named a student) [3, 4]. Recently, many deep KD variants have been proposed, e.g., self knowledge distillation (Self KD) which trains a single learner and leverages its own knowledge [2, 1], MKD with knowledge transfer between two learners [5–7], ensemble-based KD methods [8, 9], and born-again networks with knowledge distilling from multiple student generations [10]. Since we focus on training two learners, Teacher→Student KD (T2S KD) and MKD are more relevant. We briefly present them as follows and more related work is provide in Appendix A.

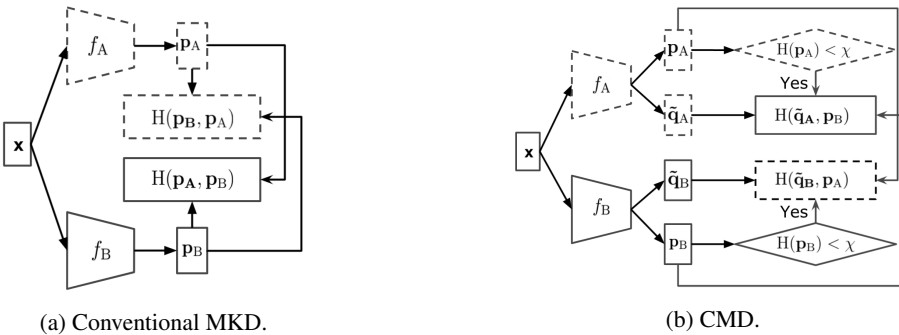

| (a) Conventional MKD. | (b) CMD. |

Figure 1: Comparison of conventional MKD and our CMD. Dotted frames represent components from model A and solid frames represent components from model B. $\mathbf{p_A}$ and $\mathbf{p_B}$ are predictions from mode A and model B, respectively. In (b), $\tilde{\mathbf{q}}_\mathbf{A}$ and $\tilde{\mathbf{q}}_\mathbf{B}$ represent the refined labels by a self distillation method, and $\chi$ is the threshold to decide whether the prediction is confident enough or not. $H(\mathbf{p})$ denotes the entropy of $\mathbf{p}$, and $H(\mathbf{q}, \mathbf{p})$ is the cross entropy loss between $\mathbf{q}$ and $\mathbf{p}$.

**T2S KD** [4] transfers knowledge from a teacher model to a student model and be formulated as:

$$L_{\text{T2SKD}}(\mathbf{q}, \mathbf{p}_s, \mathbf{p}_t) = (1 - \epsilon)H(\mathbf{q}, \mathbf{p}_s) + \epsilon D_{\text{KL}}(\mathbf{p}_t, \mathbf{p}_s), \qquad (1)$$

where $\mathbf{q}$ is the given one-hot label, $\mathbf{p}$ is the predicted distribution by a student model and $\mathbf{p}_t$ is the output of a teacher model. $H(\mathbf{q}, \mathbf{p})$ represents the cross entropy loss between target $\mathbf{q}$ and prediction $\mathbf{p}$. $D_{\text{KL}}(\mathbf{p}_t, \mathbf{p})$ denotes the Kullback–Leibler (KL) divergence of $\mathbf{p}_t$ from $\mathbf{p}$.

**MKD** [5] trains two models A and B, making them learn from each other as follows:

$$\begin{aligned} L_A(\mathbf{q}, \mathbf{p_A}, \mathbf{p_B}) &= (1 - \epsilon)H(\mathbf{q}, \mathbf{p_A}) + \epsilon D_{\text{KL}}(\mathbf{p_B}, \mathbf{p_A}) \\ L_B(\mathbf{q}, \mathbf{p_B}, \mathbf{p_A}) &= (1 - \epsilon)H(\mathbf{q}, \mathbf{p_B}) + \epsilon D_{\text{KL}}(\mathbf{p_A}, \mathbf{p_B}) \\ L_{\text{MKD}} &= L_A(\mathbf{q}, \mathbf{p_A}, \mathbf{p_B}) + L_B(\mathbf{q}, \mathbf{p_B}, \mathbf{p_A}). \end{aligned} \qquad (2)$$

## 3 Method

We design a generic knowledge selection formulation that unifies zero knowledge, all knowledge, and partial knowledge selection in a static and progressive fashion (CMD-S and CMD-P). The pseudocode of the algorithm is provided in the Appendix D.

### 3.1 Learning Objectives

To distill model B's confident knowledge into model A, we optimise A's predictions towards B's confident predictions:

$$L_{\text{B2A}} = \begin{cases} H(\tilde{\mathbf{q}}_\mathbf{B}, \mathbf{p_A}), & H(\mathbf{p_B}) < \chi, \\ 0, & H(\mathbf{p_B}) \geq \chi. \end{cases} \qquad (3)$$

We use the entropy $H(\mathbf{p_B})$ to measure the confidence of $\mathbf{p_B}$. Low entropy indicates high confidence, and vice versa [11, 12, 4, 13]. $\chi$ is a threshold to decide whether a label prediction is confident enough or not. Specifically, only when $H(\mathbf{p_B}) < \chi$, the model B's knowledge w.r.t. $\mathbf{x}$ is confident enough. $\tilde{\mathbf{q}}_\mathbf{B}$ is the model B's learning target, which can be generated by a self label correction method as it is more reliable. Note that instead of directly distilling confident predictions $\mathbf{p_B}$, we transfer targets (refined labels) $\tilde{\mathbf{q}}_\mathbf{B}$ that produce confident predictions.

Analogously, we distill model A's confident knowledge into model B:

$$L_{\text{A2B}} = \begin{cases} H(\tilde{\mathbf{q}}_\mathbf{A}, \mathbf{p_B}), & H(\mathbf{p_A}) < \chi, \\ 0, & H(\mathbf{p_A}) \geq \chi. \end{cases} \qquad (4)$$

The final loss functions for models A and B are:

$$L_A = L_{A_{\text{SelfKD}}} + L_{\text{B2A}} = \begin{cases} H(\tilde{\mathbf{q}}_\mathbf{A}, \mathbf{p_A}) + H(\tilde{\mathbf{q}}_\mathbf{B}, \mathbf{p_A}), & H(\mathbf{p_B}) < \chi, \\ H(\tilde{\mathbf{q}}_\mathbf{A}, \mathbf{p_A}), & H(\mathbf{p_B}) \geq \chi. \end{cases} \qquad (5)$$

$$L_B = L_{B_{SelfKD}} + L_{A2B} = \begin{cases} H(\tilde{q}_B, p_B) + H(\tilde{q}_A, p_B), & H(p_A) < \chi, \\ H(\tilde{q}_B, p_B), & H(p_A) \geq \chi. \end{cases} \tag{6}$$

## 3.2 A Generic Design for Knowledge Selection

As aforementioned, we use an entropy threshold $\chi$ to decide whether a piece of knowledge is certain enough or not. We design a generic formation for $\chi$ as follows:

$$\chi = \frac{H(u)}{\eta} * 2s(\frac{t}{\Gamma}-, b), \tag{7}$$

where $s(\cdot, \cdot)$ is a logistic function. $u$ is a uniform distribution, thus $H(u)$ is a constant. $t$ and $\Gamma$ denote the current epoch and the total number of epochs, respectively. For a wider unification, we make the design of Eq. (7) generic and flexible. Therefore, we use $\eta$ to control the starting point. While $b$ controls how the knowledge selection changes along with $t$. $\chi$ has two different modes:

- **Static (CMD-S)**. The confidence threshold $\chi$ is a constant when $b = 0$. Concretely, $2s(\frac{t}{\Gamma} - 0.5, 0) = 1 \to \chi = \frac{H(u)}{\eta}$. This mode covers two special cases:
  (i) One model's all knowledge is distilled into the other when $\eta \in (0, 1] \to \chi \geq H(u)$, which degrades to be the conventional MKD.
  (ii) Zero knowledge is distilled between two models when $\eta \in \{+\infty, \mathbb{R}^-\} \to \chi \leq 0$.
- **Progressive (CMD-P)**. When $b \neq 0$, $\chi$ changes as the training progresses. To make it comprehensive, $\chi$ can be either increasing or decreasing at training:
  (i) If $b > 0$, $\chi$ increases as t increases. Since the knowledge selection criteria is relaxed, more knowledge will be transferred between the two models at the later learning phase.
  (ii) On the contrary, $\chi$ gradually decreases when setting $b < 0$. This only allows knowledge with higher confidence (lower entropy) to be distilled.

## 4 Experiments

In this section, we first demonstrate that CMD is effective in robust learning against an adverse condition, i.e., label noise (Section 4.1). Then we empirically verify that CMD, as a selective MKD, outperforms prior MKD approaches for training two models collaboratively no matter whether they are of the same architecture or not (Section 4.2). We subsequently present a comprehensive ablation study and hyper-parameters analysis (Sections 4.3). Different network architectures are evaluated. For all experiments, we report the final results when the training terminates. For a more thorough comparison, we also provide an alternate self-training method called MyLC in Appendix B. More implementation details are provided in the Appendix C. The code will be released once this work is accepted.

### 4.1 CMD for Robust Learning Against Noisy Labels

**Label noise generation**    We verify the effectiveness of our proposed CMD on both synthetic and real-world label noise. For synthetic label noise, we consider symmetric noise and pair-flip noise [14]. For symmetric label noise, a sample's original label is uniformly changed to one of the other classes with a probability of noise rate $r$. The noise rates are set to 20%, 40%, 60%, and 80%. For pair-flip noise, the original label is flipped to its adjacent class with noise rates of 20% and 40%, respectively.

#### 4.1.1 The Interaction Between CMD and Self Label Correction

As shown in Tables 1 and 2, CMD, as a new selective MKD method, can be easily combined with existing self training methods as a collaborative mutual enhancer.

In Table 1, we explore to train each model using self label correction methods (LS, CP, ProselfLC [2] and MyLC). At the same time, we try four types of knowledge communication: Zero/no knowledge is distilled into the peer model and two models are trained independently; All knowledge is distilled without selection, as SyncMKD does; our proposed methods including CMD-S and CMD-P. *Vertically*, from the selective knowledge distillation perspective, we clearly observe that CMD methods (CMD-S and CMD-P) are better than "Zero" and "All" consistently no matter how each model is trained. This empirically demonstrates that selecting confident knowledge for distillation is better. In addition,

Table 2: Results on CIFAR-100 clean test set. All methods use ResNet34 as the network architecture. The top results of each column are bolded.

| Method | Pair-flip label noise | | Symmetric label noise | | Clean |
| --- | --- | --- | --- | --- | --- |
| | 20% | 40% | 20% | 40% | |
| CE | 63.52 | 45.40 | 63.31 | 47.20 | 75.58 |
| LS | 65.15 | 50.02 | 67.45 | 51.53 | 76.33 |
| CP | 64.97 | 49.01 | 65.97 | 51.09 | 75.29 |
| Boot-soft | 64.04 | 48.85 | 63.25 | 48.41 | 75.37 |
| ProSelfLC | 74.13 | 69.49 | 71.49 | 64.07 | 75.73 |
| CMD-S+ProselfLC | 75.68 | 74.22 | 72.11 | 67.26 | 76.25 |
| CMD-P+ProselfLC | 75.76 | 74.55 | 72.58 | 68.29 | **77.32** |
| MyLC | 73.12 | 62.29 | 71.04 | 65.04 | 75.20 |
| CMD-S+MyLC | 75.39 | 74.32 | 72.20 | 68.45 | 75.92 |
| CMD-P+MyLC | **75.89** | **74.72** | **73.22** | **69.09** | 76.42 |

Table 3: Recent approaches for label noise are compared. All methods apply ResNet50 as the network architecture. For Food-101, we use a ResNet50 pre-trained on ImageNet. For Webvision, we follow the "Mini" setting in [15–18]. The top results of each column are bolded.

| Method | CIFAR-100 | | | | Real-world noise | |
| --- | --- | --- | --- | --- | --- | --- |
| | Pair-flip label noise | | Symmetric label noise | | Food-101 | Webvision (Mini) |
| | 20% | 40% | 20% | 40% | ∼20% | ∼50% |
| CE | 64.10 | 52.77 | 63.93 | 56.82 | 84.03 | 57.34 |
| GCE [19] | 62.32 | 55.03 | 65.62 | 57.97 | 84.96 | 55.62 |
| Co-teaching [14] | 58.11 | 48.46 | 61.47 | 53.44 | 83.73 | 61.22 |
| Co-teaching+ [20] | 56.31 | 38.03 | 64.13 | 55.92 | 76.89 | 33.26 |
| Joint [21] | 67.35 | 52.22 | 54.88 | 45.64 | 83.10 | 47.60 |
| Forward [22] | 58.37 | 39.82 | 66.12 | 59.45 | 85.52 | 56.33 |
| MentorNet [16] | 54.73 | 45.31 | 57.27 | 49.01 | 81.25 | 57.66 |
| T-revision [23] | 62.69 | 52.31 | 64.67 | 57.15 | 85.97 | 60.58 |
| DMI [24] | 58.77 | 42.89 | 62.77 | 57.42 | 85.52 | 56.93 |
| S2E [25] | 58.21 | 41.74 | 64.21 | 43.12 | 84.97 | 54.33 |
| APL [18] | 59.77 | 53.25 | 59.37 | 51.03 | 82.17 | 61.27 |
| CDR [15] | 71.93 | 56.94 | 68.68 | 62.72 | 86.36 | 61.85 |
| ProSelfLC [2] | 73.11 | 69.49 | 71.17 | 60.38 | 86.97 | 62.40 |
| CMD-P+ProselfLC | **75.16** | 73.36 | **73.25** | 64.09 | 87.54 | 67.40 |
| MyLC | 72.25 | 70.84 | 69.92 | 62.80 | 86.70 | 64.44 |
| CMD-P+MyLC | 74.38 | **73.86** | 72.23 | **64.30** | **87.60** | **67.48** |

CMD-P is slightly better than CMD-S, mainly due to the fact that a model's knowledge upgrades and becomes confident as the training progresses.

Table 2 is an extension of Table 1. Results of different noise types and rates are present. Since ProSelfLC and MyLC always performs better than the other approaches, therefore we only apply CMD over them to explore how much CMD can enhance stronger baselines.

### 4.1.2 Comparison with Learning with Noisy Labels Methods

In this subsection, our objective is to compare with recent methods for addressing label noise. For simplicity, we only train CMD-P together with ProSelfLC and MyLC, which are demonstrated to be the best in Section 4.1.1. Table 3 (CIFAR-100) shows results of training ResNet50 on CIFAR-100. CMD-P+ProSelfLC and CMD-P+MyLC outperform all the recent label-noise-oriented methods under both pair-flip and symmetric noisy labels. Notably, their improvements are more significant when noise rate rises. We also presents the results on two real-world noisy datasets, Webvision and Food-101 in Table 3. For Webvision, we follow the "Mini" setting in [16]. The first 50 classes of the Google resized image subset is treated as training set and evaluate the trained networks on the same 50 classes on the ILSVRC12 validation set. The results of CMD-P+ProSelfLC and CMD-P+MyLC

Table 4: The performance of CMD under different settings, two distinct architectures, and the same architectures. CMD+MyLC outperforms other MKD methods.

| Method | | Difference | | Same |
| | | ResNet18 | ShufflenetV2 | ResNet34 |
|---|---|---|---|---|
| Baseline | CE | 50.63 | 44.06 | 47.20 |
| Self KD | Tf-KD$_{reg}$ [1] | 51.05 | 44.70 | 47.39 |
| | ProselfLC [2] | 58.51 | 58.89 | 64.07 |
| | MyLC | 55.94 | 61.21 | 65.04 |
| MKD | MKD [5] | 60.38 | 47.72 | 51.42 |
| | KDCL [8] | 55.45 | 46.10 | 51.20 |
| | CMD+MyLC | **68.10** | **64.37** | **69.09** |

Table 5: The results of CMD-S with different $\eta$. We train on CIFAR-100 using ResNet-34.

| CMD-S | Symmetric label noise | | | |
| | 20% | 40% | 60% | 80% |
|---|---|---|---|---|
| H(u) ($\eta = 1$) | 70.37 | 59.26 | 36.18 | 16.17 |
| 1/2 H(u) ($\eta = 2$) | 72.11 | 65.04 | 46.15 | 18.62 |
| 1/3 H(u) ($\eta = 3$) | 72.83 | 66.42 | 51.34 | 19.84 |
| 1/4 H(u) ($\eta = 4$) | **73.25** | **67.26** | **54.34** | **22.45** |

are around **5-6%** higher than the latest methods including Co-teaching, APL, CDR, and ProselfLC. Due to the increased difficulty of Food-101, the performance gap across techniques is narrower. CMD-P+ProSelfLC and CMD-P+MyLC regularly outperform all compared algorithms.

## 4.2 Comparing with Recent MKD Methods

In Table 4, we present the results of the baseline CE, self KD methods (Tf-KD$_{reg}$ [1], ProselfLC and MyLC), and mutual distillation algorithms (MKD, KDCL, CMD-P+ProSelfLC, and CMD-P+MyLC) under noisy scenarios. For self KD methods, we train each model individually (i.e., without mutual distillation) while for MKD methods, we train them together (i.e., with mutual distillation).

- **MKD for two networks of the same architecture.** In Table 4 (same), CMD-P+MyLC achieves **17%-18%** absolute improvement compared to MKD and KDCL. All experiments are trained for 100 epoch.

- **MKD for two networks of different architectures.** In Table 4 (difference), we demonstrate CMD's effectiveness for training two different networks, ResNet18 and ShuffleNetV2. CMD improves MyLC for around 3% for ResNet18 and 1-3% for ShuffleNetV2. Each experiment is trained for 200 epoch.

## 4.3 Hyper-parameters Analysis

### 4.3.1 Analysis of $b$

Mathematically, according to section 3.2, $b$ decides how the knowledge selection threshold changes along with the training epoch $t$. In Figure 2a, we fix $\eta = 2$ and study the effect of $b$ under different noise rates. We observe that the accuracy increases as $b$ decreases for all noise rates. The trend becomes more obvious as the noise rate increases. This empirically verifies the effectiveness of confident knowledge selection again. Furthermore, progressively increasing the confidence criterion leads to better performance. In Figure 2b, we further study $b$ under different $\eta$. The accuracy keeps increasing as $b$ decreases for all $\eta$. Additionally, the trend is more significant when $\eta$ becomes smaller.

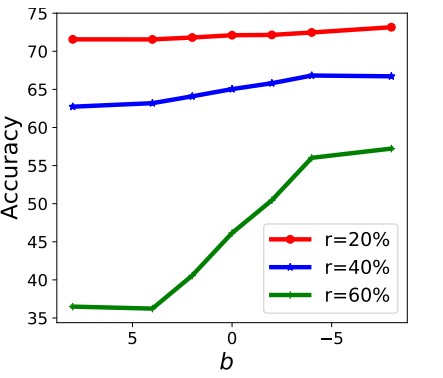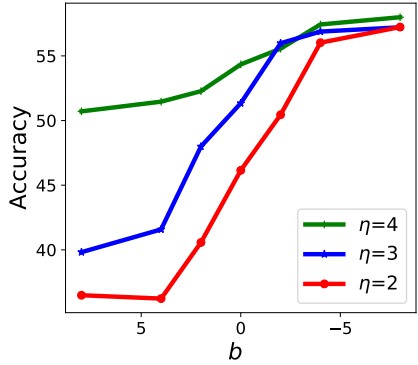

(a) Under different noise rates with $\eta = 2$     (b) Under different $\eta$ with noise rate $r = 60\%$

Figure 2: Analysis of $b$ under CIFAR-100.

### 4.3.2 Analysis of $\eta$

As presented in section 3.2, $\eta$ is a parameter to linearly scale the knowledge selection criteria. To study $\eta$, we first analyze the static mode. Table 5 shows the results of CMD-S with different $\eta$. We can see that a lower threshold (i.e., larger $\eta$) has higher accuracy for all noise rates. This further demonstrates the effectiveness of distilling more confident knowledge. We then analyse the dynamic mode. In Figure 2b, the green line ($\eta = 4$) has the highest accuracy for most $b$ values. Overall, the blue line ($\eta = 3$) is the second best, while the red line ($\eta = 2$) has the lowest accuracy. Therefore, we conclude that a smaller $\eta$ is better in both static and progressive modes.

## 5 Conclusion

We are investigating knowledge selection in MKD and proposing an unified framework for knowledge selection called CMD. CMD improves MKD by distilling only confident knowledge to the peer model. Extensive experiments illustrate the effectiveness of CMD empirically. In addition, our suggested CMD outperforms comparable MKD algorithms in the presence of label noise and achieves competitive performance in clean circumstances.

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

# A  Related Work

## A.1  Ensemble-based and Feature-map-based KD Methods

Knowledge Distillation via Collaborative Learning (KDCL) [8] treats all models as students, while the teacher model is an ensemble of all students. Peer Collaborative Learning (PCL)[9] assembles multiple subnetworks as a teacher model. FFL[26] integrates feature representation of multiple models and AFD[27] transfers prediction and feature-map knowledge together.

## A.2  Learning with Noisy Labels

We compare the recent methods for learning with noisy labels. For example, selecting confident samples, *Co-teaching* [14] and *Co-teaching+* [20] maintain two identical networks simultaneously and transferring small-loss instances to the peer model; *MentorNet* [16] provides a curriculum for StudentNet to focus on the examples with likely-correct labels. *Joint* [21] and *Forward* [22] correct training loss through the calculation of the noise transition matrix. Sample reweighting, e.g., *T-revision* [23] reweights samples based on their significance. Designing robust loss function, *DMI* [24] introduces an information-theoretic loss function, and *APL* [18] combine two robust loss functions that mutually boost each other. Early stopping, *CDR* [15] reduces the side effect of noisy labels before early stopping. Label correction, Joint [21] and ProselfLC [2] refine noisy labels by confident predictions. Label correction is commonly employed in settings with label noise, but it can also be used in clean situations for regularization. More information in Section A.3

## A.3  Label Correction

As mentioned in [1], the learning target modification is to replace a one-hot label representation by its convex combination with a predicted distribution $\tilde{\mathbf{p}}$: $\tilde{\mathbf{q}} = (1 - \epsilon)\mathbf{q} + \epsilon\tilde{\mathbf{p}}$. $\epsilon$ measures how much we trust the prediction, and it can be fixed in Label smoothing(LS) [12], Confidence penalty (CP) [11], Boot-soft [28], Joint-soft [21], or adaptive by training time e.g., [2] and [1]. In Appendix B, we also present an alternative label correction approach, MyLC, in which *epsilon* is updated by model confidence. $\tilde{\mathbf{p}}$ can originate from various sources, such as uniform distributions, a current model, a model that has been pretrained, etc. By adding a uniform distribution, for example, LS reduces the confidence in annotated label. CP reduces the credibility of annotated labels by penalizing high confidence predictions. By incorporating a related prediction, Boot-soft, Tf-KD, and MyLC refine the learning target.

# B  MyLC: An Alternative for Label Correction

MyLC is designed for demonstrating the effectiveness and extensiveness of CMD, which serves as an alternative to label correction methods. Note that MyLC is different from ProselfLC methods in terms of working principle. Furthermore, MyLC solves a significant drawback of ProselfLC that the model always has to be trained from scratch, since ProselfLC relies on training time. MyLC is obviously more suitable if we want to do fine-tuning or incremental learning tasks based on pretrained models. Specifically, without considering training time, MyLC defines the global model confidence according to a model's predictive confidence w.r.t. all samples and is computed as follows:

$$g(r) = s(r - \rho, b_1), \quad \text{where } r = 1 - \frac{\sum_{i=1}^{n} \mathrm{H}(\mathbf{p}_i)}{n * \mathrm{H}(\mathbf{u})}. \tag{8}$$

$s(\lambda, b_1) = 1/(1 + \exp(-\lambda \times b_1))$ is a logistic function, where $b_1$ is a hyperparameter for controlling the smoothness of $h$. This is widely used in semi-supervised learning[29, 30] and label noise learning [2]. $r$ represents a model's overall certainty of all examples. A higher $r$ implies that a model is more reliable. Intuitively, if $r$ is higher than a threshold $\rho$, we should assign more trust to the model. We simply set $\rho = 0.5$ in all our experiments. Consequently, Consequently, $\epsilon = g(r) \times l(\mathbf{p})$. And the loss becomes:

$$L_{\mathrm{MyLC}} = \mathrm{H}(\tilde{\mathbf{q}}_{\mathrm{MyLC}}, \mathbf{p}), \text{where } \tilde{\mathbf{q}}_{\mathrm{MyLC}} = (\mathbf{1} - \epsilon)\mathbf{q} + \epsilon\mathbf{p}. \tag{9}$$

## C   Implementation Details

### C.1   Datasets and Data Augmentation

- CIFAR100 [31] has 50,000 training images and 10,000 test images of 100 classes. The image size is $32 \times 32 \times 3$. Simple data augmentation is applied following [32], i.e., we pad 4 pixels on every side of the image and then randomly crop it with a size of $32 \times 32$.

- Food-101 [33] has 75,750 images of 101 classes. The training set contains real-world noisy labels. In the test set, there are 25,250 images with clean labels. For data augmentation, training images are randomly cropped with a size of $224 \times 224$.

- Webvision [16] has 2.4 million images crawled from the websites using the 1,000 concepts in ImageNet ILSVRC12 [34]. For data augmentation, we first resize the training images to $320 \times 320$ and then randomly cropped with a size of $299 \times 299$.

### C.2   Training Details

- On CIFAR100, we train on 90% training data (corrupted in synthetic cases) and use 10% clean training data as a validation set to search hyperparameters, e.g., $b_1$, $b_2$. Finally, we retrain a model on the entire training data and report its accuracy on the test data for a fair comparison. We train CIFAR100 on three net architectures including ResNet34, ResNet50, ResNet18 and ShuffleNetV2. For ResNet34, the initial learning rate is 0.1 and then divided by 10 at the 50th and 80th epoch, respectively. The number of total epochs is 100. For ShuffleNetV2 and ResNet28, the initial learning rate is 0.1 and then divided by 5 at the 60th, 120th, and 160th epoch, respectively. We train 200 epochs in total. For all the training, we use an SGD optimizer with a momentum of 0.9, a weight decay of 5e-4, and a batch size of 128. For ResNet50, for a fair comparison, we use the same training settings as [15].

- On Food-101, we also separate the training data into two parts, 90% for training and 10% for validation. We use the validation set to search hyper-parameters. Finally, we report its accuracy on the clean test data. We train ResNet50 (initialised by a pretrained model on ImageNet) using a batch size of 32, due to GPU memory limitation. And we use the SGD as an optimizer with a momentum of 0.9, and a weight decay of 5e-4. The learning rate starts at 0.01 and then is divided by 10 at the 50th and 80th epoch, respectively in total 100 epochs.

- On Webvision, we follow the "Mini" setting in [16]. We take the first 50 classes of the Google resized image subset as the training set and the same 50 classes of the ILSVRC12 validation set as the test set and apply inception-resnet v2 [35] as training architecture with batch size of 32. We use SGD as an optimizer with a momentum of 0.9, and a weight decay of 5e-4. The learning rate starts at 0.01 and then is divided by 10 in each epoch after the 40th epoch with a total number of 80 epochs.

All models are trained on multiple 2080 Ti GPUs between 2 and 4, which is adjusted according to model size and batch size.

# D    The Core Implementation of CMD Using PyTorch

```python
class CMDWithLoss(nn.Module):
    def __init__(self):
        super(CMDWithLoss, self).__init__()

    def forward(self, qA, qB, pA, pB, threshold):
        # qA, corrected label from model A
        # qB, corrected label from model B
        # pA, knowledge from model A
        # pB, knowledge from model B

        # calculate the entropy of pA
        hpA = torch.sum(-pA * torch.log(pA + 1e-6), 1)
        # calculate the entropy of pB
        hpB = torch.sum(-pB * torch.log(pB + 1e-6), 1)

        threshold_l = threshold * torch.ones(len(hpA)).cuda()

        # select the low entropy sample from model B
        indexA = (hpB < threshold_l).nonzero()
        # select the low entropy sample from model A
        indexB = (hpA < threshold_l).nonzero()

        # distill knowledge from model B to model A
        lossB2A = torch.sum(qB[indexA].squeeze(1) * \
                  (-torch.log(pA[indexA].squeeze(1) + 1e-6)), 1)
        # distill knowledge from model A to model B
        lossA2B = torch.sum(qA[indexB].squeeze(1) * \
                  (-torch.log(pB[indexB].squeeze(1) + 1e-6)), 1)

        lossB2A = sum(lossB2A) / len(hpA)
        lossA2B = sum(lossA2B) / len(hpB)

        return lossB2A, lossA2B
```

