# OpenReview forum: "Not All Knowledge Is Created Equal: Mutual Distillation of Confident Knowledge"
_NeurIPS.cc/2022/Workshop/TSRML — TSRML2022_

### Official Review · Reviewer_wMf1 · 2022-10-21
**An intuitive work on adaptive knowledge sharing for distillation.**

**Overall Recommendation:** Given the effectiveness of the propos…
**Overall Rating:** 7

**Summary:**

This work aims at enabling adaptive selection of knowledge sharing in Mutual Knowledge Distillation (MKD). The proposed method, termed Confident knowledge selection followed by Mutual Distillation (CMD) is based on the idea that only confident knowledge should be shared for distillation, and thus uses a thresholding mechanism to filter out those that lack sufficient certainty, measure by entropy. Enabling static or adaptive/progressive adjustment of this threshold gives a general framework that subsumes zero as well as "all" knowledge sharing as special cases.

**Strengths:**

The problem is interesting, and proposed approach is simple yet intuitive. It is also applied on setting with noisy labels where selecting confident knowledge is expected to help, and it shows good numerical results. The paper is well written and easy to follow.

**Weaknesses:**

In terms of novelty, the work is limited, however the proposed method is concise and intuitive.

**Review Confidence:**

3: The reviewer is fairly confident that the evaluation is correct

---

### Decision · Program_Chairs · 2022-10-23

Accept